# Performance of Contrast-Enhanced Ultrasound in Thyroid Nodules: Review of Current State and Future Perspectives

**DOI:** 10.3390/cancers13215469

**Published:** 2021-10-30

**Authors:** Maija Radzina, Madara Ratniece, Davis Simanis Putrins, Laura Saule, Vito Cantisani

**Affiliations:** 1Radiology Research Laboratory, Riga Stradins University, LV-1007 Riga, Latvia; ratniece.madara@gmail.com (M.R.); laura.saule@rsu.lv (L.S.); 2Medical Faculty, University of Latvia, LV-1004 Riga, Latvia; davisputrins@gmail.com; 3Diagnostic Radiology Institute, Paula Stradina Clinical University Hospital, LV-1002 Riga, Latvia; 4Department of Radiological, Anatomopathological and Oncological Sciences, Sapienza University of Rome, 00100 Rome, Italy; vito.cantisani@uniroma1.it

**Keywords:** contrast-enhanced ultrasound (CEUS), thyroid nodules, thyroid cancer, papillary thyroid cancer, follicular thyroid cancer

## Abstract

**Simple Summary:**

Ultrasound has been used as baseline imaging for thyroid nodules for decades; nevertheless, no single feature is sensitive or specific enough to exclude or confirm thyroid malignancy. Therefore, clinical practice and research still focus on less invasive diagnostic patterns to reduce unnecessary fine-needle aspiration biopsies or surgery. The main advantage of CEUS is the ability to assess the sequence and intensity of vascular perfusion and hemodynamics in the thyroid nodule, thus providing real-time characterization of nodule features, and considered a valuable new approach in the determination of benign vs. malignant nodules. In addition, contrast agents used in CEUS can help to guide treatment planning for minimally invasive procedures (e.g., ablation) and to provide accurate follow-up imaging to assess treatment efficacy in both benign and malignant nodules and associated lymph nodes. Examination protocol has almost reached standardization, although there are numerous controversies reported about the interpretation of qualitative and quantitative patterns that would require a systematic approach. This literature and current state review of CEUS in thyroid nodules address the existing concepts and highlights of the future perspectives.

**Abstract:**

Ultrasound has been established as a baseline imaging technique for thyroid nodules. The main advantage of adding CEUS is the ability to assess the sequence and intensity of vascular perfusion and hemodynamics in the thyroid nodule, thus providing real-time characterization of nodule features, considered a valuable new approach in the determination of benign vs. malignant nodules. Original studies, reviews and six meta-analyses were included in this article. A total of 624 studies were retrieved, and 107 were included in the study. As recognized for thyroid nodule malignancy risk stratification by US, for acceptable accuracy in malignancy a combination of several CEUS parameters should be applied: hypo-enhancement, heterogeneous, peripheral irregular enhancement in combination with internal enhancement patterns, and slow wash-in and wash-out curve lower than in normal thyroid tissue. In contrast, homogeneous, intense enhancement with smooth rim enhancement and “fast-in and slow-out” are indicative of the benignity of the thyroid nodule. Even though overlapping features require standardization, with further research, CEUS may achieve reliable performance in detecting or excluding thyroid cancer. It can also play an operative role in guiding ablation procedures of benign and malignant thyroid nodules and metastatic lymph nodes, and providing accurate follow-up imaging to assess treatment efficacy.

## 1. Introduction

Thyroid nodules are defined as discrete lesions within the thyroid gland that radiologically differ from the surrounding thyroid parenchyma [1]. With the increasingly widespread availability and use of diagnostic imaging modalities such as ultrasound (US), computed tomography (CT), magnetic resonance imaging (MRI) and positron emission tomography (PET), thyroid nodules have been more frequently identified [2]. The main goal of diagnostics in thyroid nodules is to exclude thyroid cancer by differentiating benign from malignant nodules [3,4]. The prevalence of thyroid nodules detected by ultrasound has been reported up to 50% in the general population [5], with a malignancy rate of 5–15% [1,6]. Despite the relatively high incidence, thyroid cancer mortality has remained rather stable over a longer period, with the ten-year relative survival rate reported to be greater than 90% [7]. The most common type of thyroid cancer is papillary carcinoma, accounting for 85% of all thyroid malignancies [8].

Due to the high incidence of thyroid nodules, a noninvasive and relatively easily accessible imaging modality is ultrasound, which has proven to be highly sensitive in the detection and characterization of different thyroid nodules [9], and is recommended as the first-line modality to be performed in all patients with suspected thyroid nodules [1,10]. Therefore this review will be dedicated to US. The US appearance of a nodule is crucial in the management of a patient, most importantly when deciding whether further investigation with fine-needle aspiration (FNAB) or routine follow-up is necessary [11]. By maximizing the detection of clinically relevant thyroid lesions and decreasing FNAB of benign nodules, the best patient outcomes and cost-effectiveness can be achieved due to reduced overdiagnosis and overtreatment [10].

To date, well known and guideline-approved malignancy US features have been defined, including solid composition, hypoechoic appearance, calcifications, ill-defined margins, “taller than wide” configuration, and a lack of “halo” [12]. However, no single feature is both sensitive and specific enough to exclude or confirm thyroid malignancy [10,13], for example calcifications can be seen in up to 40.2% of malignant and 22.2% of benign nodules as reported by Kim et al. [14]. In addition to the conventional B-mode imaging, color Doppler studies and methods for examining microvascular flow are also utilized in the evaluation of thyroid nodules, with marked hypervascularity being a commonly accepted indicator of potential malignancy in certain cases, although many guidelines recommend not to use vascularization to predict malignancy [15,16]. As such Doppler US is not recommended as a routine method for US malignancy risk stratification in the EU-TIRADS guidelines [11], and is also not considered to be a reliable indicator of malignancy in other guidelines, such as the ACR-TIRADS [17].

Ultrasound is used as the basis for several thyroid nodule risk stratification systems proposed by the American Thyroid Association (ATA), American College of Radiologists (ACR), European Thyroid Association (EU-TIRADS), and others, which are in place to increase the diagnostic confidence of thyroid nodule imaging. A widely used thyroid imaging reporting and data system (TI-RADS) has achieved good clinical value by improving the diagnostic accuracy and reducing unnecessary biopsies, and overall has affected the management of thyroid nodules [18]. However, TI-RADS staging largely depends on the operator, and several specific imaging features are not universally accepted in evaluation [10,19]. Furthermore, features of atypical benign and malignant nodules, especially TI-RADS 3 and 4 category, may overlap the routine US and even FNAB appears to prove half of all biopsied nodules as benign [20], and in up to one third of cases the results of cytology are inconclusive [2]. Therefore, problem-solving modalities of US have been introduced in the past two decades such as US elastography (strain elastography and shear-wave elastography [21,22] and contrast-enhanced ultrasound (CEUS), which have expanded the ability of conventional US.

The aim of this review was to analyze different aspects of CEUS in thyroid nodules, including technical considerations, qualitative and quantitative analysis of the following benign and malignant thyroid lesions: adenoma, goiter, thyroiditis, lymphoma, papillary, follicular, medullary and anaplastic thyroid cancer, while also providing an overview on its role in pre- and post-treatment nodule and specific lymph node local assessment.

## 2. Materials and Methods

In the present paper a comprehensive literature search of PubMed, Google Scholar and Scopus databases was conducted with MESH terms: “CEUS or Contrast-Enhanced Ultrasonography” and “thyroid nodule or thyroid cancer”; to investigate the role of CEUS in evaluation of efficacy of performed treatment on thyroid nodules and nodal involvement the MESH terms “CEUS or Contrast-Enhanced Ultrasonography” and “thyroid nodule or thyroid cancer” and “after treatment” were used. The search was updated from 2010 until June of 2021 and references of the retrieved articles were explored. Original studies, reviews and 6 meta-analyses were included in this article. A total of 624 studies were retrieved, and 107 were included in the study. To avoid bias, only studies with histological reference as the gold standard were included and all of the MESH terms should have been present in the titles or abstracts.

## 3. Results

### 3.1. The Technique of Contrast-Enhanced Ultrasound in Thyroid Imaging

The main advantage of CEUS is the ability to assess the sequence and intensity of vascular perfusion and hemodynamics in a thyroid nodule, therefore providing real-time characterization of nodule features after an intravenous bolus injection of a microbubble contrast agent (CA) [23]. Another advantage of CEUS is a lack of contrast media adverse effects or nephrotoxicity (1:10,000 vs. 1–12:100 of iodinated contrast agents) [24]. Adverse effects from CEUS microbubble contrast agents are shown to be sparse, which could be related to clearly intravascular distribution, and their overall safety is very reliable, especially when compared to the potential side effects of CT and MR and the use of their associated contrast agents [25,26].

In CEUS, both qualitative and quantitative evaluation is possible for thyroid nodules [27], and it has been shown by Trimboli et al. to have a good CEUS polled sensitivity of 85% across different studies and a specificity of 82%; in the recent meta-analysis, positive and negative predictive values were 83% and 85%, respectively [28].

A typical protocol for thyroid nodules CEUS examination includes low mechanical index (MI < 0.10) and the focus zone should be placed at the lower portion of the FOV [29] preferably including the entire nodule and surrounding thyroid tissue in the longitudinal plane. When the conventional B-mode image is properly adjusted and the CEUS mode activated, a microbubble contrast agent is injected as an intravenous bolus, generally with a dose of around 1.0–2.0 mL, followed by 5–10 mL of saline; however, specific amounts of CA and saline vary between operators and institutions [30,31,32,33]. A CEUS timer is started simultaneously with the injection of the contrast agent, and cine-clips of the scanning are stored digitally as raw data for 2–3 min before being processed, but the length of examination differs between sources and institutions, with most authors choosing a time period of at least 2 min [34,35]. Only one nodule can be evaluated for each injection of contrast agent.

First evaluation includes qualitative analysis—the presence of enhancement, washout and comparison to the normal thyroid parenchyma. After acquiring raw data, a more detailed post-processing of the nodule can be carried out using dedicated software, including quantitative analysis. The mainstay and basis of qualitative analysis include evaluation of contrast medium entry time in the nodule and peak enhancement, with contrast intensity being higher, lower or approximately identical to surrounding parenchyma (hyper-, hypo- or iso-enhancement, respectively), or absent enhancement [36]. The pattern is usually defined depending on the dynamics of contrast bubbles within the nodule: concentric in the case of centripetal enhancement (from the periphery) and centrifugal enhancement (towards the periphery). Non-concentric enhancement patterns include diffuse and eccentric enhancement when the contrast does not exhibit a particular directionality, and peripheral rim enhancement) [37,38,39]. Other qualitative indicators include contrast uptake homogeneity, with full enhancement, regardless of the enhancement degree, whereas heterogeneous nodules contain intra-nodular areas with various levels of enhancement; nodule borders (clearly differentiated from the surrounding parenchyma, or unclear); morphology (shape and regularity of form); and nodule size [40]. After contrast uptake, the contrast wash-out is observed: the timing of the beginning of washout and speed of this process relative to the surrounding parenchyma [41].

CEUS quantitative analysis is operator dependent: during the post-processing, the manually drawn region of interest (ROI) is placed within the nodule and surrounding parenchyma, subsequently generating variable color-coded curves, and most of the following quantitative parameters are automatically calculated and used for the analysis of the rise time, time to peak (time until peak intensity is reached), wash-in slope, peak intensity, mean transit time (intensity values are higher than the mean), and area under the time-intensity curve: all of these parameters can be used to characterize nodules and have been studied by multiple authors [29,37,42].

Despite the wide applications and interpretation possibilities provided by CEUS, no single feature is sensitive or specific enough for the determination of malignancy; moreover, there are no unified standards for quantitative and qualitative studies [27,28,43]. CEUS also suffers from various limitations that have contributed to the delay of its implementation in routine clinical practice: firstly technical, secondly interpretative, and finally economical. From the technical point of view, microbubble contrast agents last only 5–10 minutes, which shortens the time allowed for investigation, and some other tradeoffs have to be made regarding image quality in order to increase the lifespan of microbubbles, mainly lowering the MI, which simultaneously increases image noise [44,45,46]; in addition, the equipment requires specific software to perform US with low mechanical index, and together with extra contrast media costs they bring added expenses. CEUS is also a minimally-invasive diagnostic manipulation, although not more so than other methods of investigation which apply intravenous contrast media and are carried out routinely on a mass scale (e.g., computed tomography, magnetic resonance imaging) with significantly lower adverse reaction rates for CEUS, described above. Other authors also point out that CEUS depends on the experience and visual interpretation of the operator [47,48], especially in qualitative assessment. CEUS examination requires extended examination time, assistance of support medical personnel, post-procedural patient observation and, as it has not yet been approved for thyroid in the international guidelines, is not reimbursed in many countries. All of the above mentioned reasons may limit the wide use of CEUS and decrease its availability in certain cases.

### 3.2. CEUS of Thyroid Nodules: Benign vs. Malignant

Histologically normal thyroid parenchyma is rich in micro-vessels and therefore shows a rapid uniform enhancement after the administration of CA. Thyroid nodules, however, have a different vascularization pattern, therefore a different presentation on CEUS [49]. It has been reported that thyroid cancer cells secrete cytokines to stimulate angiogenesis, therefore increasing vascularization and causing distorted vessel distribution or arteriovenous fistula [50].

#### 3.2.1. Qualitative Analysis of CEUS Enhancement Patterns

Enhancement patterns for CEUS within thyroid nodules are insufficient for the diagnosis of thyroid carcinoma, although several patterns have been described [39]. Some studies classify CEUS enhancement patterns as low-enhancing, iso-enhancing and high enhancement, with low enhancement most suggestive of malignancy and sensitivity, with specificity and accuracy of 82%, 85% and 84%, respectively [51,52,53,54,55,56]. In a study conducted by Zhang et al. [57] four contrast enhancement patterns were described as: homogenous, heterogeneous, ring-enhancement and no enhancement. In this study, there was a significant difference between benign and malignant nodules with a *p* value of <0.001. For benign nodules, ring enhancement was seen in 83% of cases, homogenous and heterogenous in 7.5%, and no enhancement in 1.9%. As for malignant nodules, 88.2% showed a heterogeneous enhancement, ring enhancement was observed in 5.9%, and homogenous enhancement also in 5.9% of the cases. Ring enhancement correlated with a benign disease, with a sensitivity and specificity of 83% and 94.1%, respectively. However, heterogeneous enhancement correlated with a malignant disease with sensitivity and specificity of 88.2% and 92.5%, respectively [57]. Most malignant nodules contain areas of fibrosis, calcification, or focal necrosis, which may cause heterogeneous enhancement.

In another similar study by Zhang et al. [39], 120 nodules were characterized using CEUS in which peripheral and internal enhancement patterns were determined. In contrast to previous statements, it was concluded that peripheral irregular ring enhancement pattern on CEUS detects malignancy and improves the diagnostic accuracy of CEUS in combination with internal enhancement patterns (sensitivity 97.6%, specificity 98.7%). Interestingly enough, the size of nodules with regular peripheral enhancing rings was significantly larger than with other types of peripheral rings [39], suggesting that the type of peripheral ring observed might be related to the size of the thyroid nodule.

In a 2016 study by Zhang et al. [9], 157 thyroid lesions were analyzed with CEUS. There was a statistically significant difference for the presence of peripheral ring enhancement and different enhancement patterns in benign and malignant thyroid nodules. Most malignant nodules (70.37%) were found to have a low enhancement, and the sensitivity, specificity and accuracy of this CEUS feature to diagnose thyroid cancer were 84.15%, 65.33% and 75.16%, respectively. The irregular peripheral ring pattern on CEUS had reached a sensitivity of 100%, specificity 94.12% and accuracy of 95% for diagnosing malignancy. In this study, the pattern of iso-enhancement with focal low-enhancing areas was also commonly associated with malignancy, with low-enhancing areas corresponding to interstitial fibrosis, but enhancing areas to malignant cells. Interestingly, the misdiagnosis rate with the conventional US was 57.33%, but only 34.67% for CEUS. This study also suggested the size of the lesion impacts the enhancement pattern on CEUS with small malignant lesions more often presenting as low enhancing, possibly due to an immature vascular network in microcarcinomas [9].

Pang et al. conducted a study in which regression analysis of CEUS for differentiating benign vs. malignant nodules showcased that the hypo-enhancement pattern was highly specific for malignancy [3]. The main reason that thyroid malignant tumors show a low blood supply is related to the complex neovascularization—once the growth outweighs neovascularization, necrosis and embolus formation within the tumor leads to hypo-enhancement on CEUS.

A different approach was taken in a study by Wu et al. where 229 lesions were analyzed with the conventional US and CEUS and divided into enhancement and non-enhancement groups as well as divided into two groups of different sizes, <10 mm and >10 mm. Five indicators were analyzed: arrival time, mode of entrance, echo intensity, homogeneity, and washout time. Within the subgroup of <10 mm there was a statistically significant difference between benign and malignant thyroid nodules for arrival time, mode of entrance and washout time; however, all five [30] indicators showed a statistically significant difference within the subgroup of >10 mm and the total group. The specificity for previously mentioned indicators ranged from 90–96% and diagnostic accuracy between 75–82%. As for the non-enhancement pattern, sensitivity and NPV were 95.5% and 95.8%, respectively [55].

In a recent study by Zhao et al. in 2018 the highest accuracy of 94.02%, sensitivity of 94.74% and specificity of 93.33% was achieved using TI-RADS in combination with CEUS for differentiation of malignant vs. benign nodules. The most prominent features of nodules pointing to malignancy on CEUS were low enhancement and rim-like enhancement [58]. In contrast, other authors reported rim-enhancement as a pattern for benign nodules [30].

Several studies suggest that using CEUS and TI-RADS together can help achieve the highest diagnostic accuracy [35]; for example, in a study by Zhang et al. the diagnostic accuracy of CEUS alone was 90%, for TI-RADS 90.3% and for the combination of both 96% [59]. Some studies report in addition that using conventional US, CEUS and real time elastography increases the sensitivity and specificity of all three methods, but elastography has often been reported as the most valuable tool [49,60,61]. In a study by Deng et al. the combination of US, CEUS and acoustic radiation force impulse (ARFI) markedly improved the diagnostic accuracy when compared to either combination of two modalities [62] and, as Sui et al. reported, the accuracy of CEUS alone was 85.32% vs. CEUS and strain elastography at 95.41% [63] and, according to the recent results [64], the SWE and TIRADS combination showed improved specificity compared with the TIRADS alone (0.917 vs. 0.896), suggesting that the combination method may be valuable in reducing unnecessary FNAB in certain patients. We have to keep in mind that several benign entities may show stiff patterns (e.g., Hashimoto and Riedel’s thyroiditis) and malignancy may appear as soft lesions (e.g., follicular adenoma, carcinoma, Hurthle cell neoplasia) [65].

#### 3.2.2. Quantitative Analysis of Thyroid Nodules in CEUS

Wang et al. conducted a study using CEUS on 135 patients with histologically proven thyroid nodules. Binary logistic regression indicated several features suggesting malignancy: slow wash-in, heterogeneous enhancement, ill-defined enhancement border and fast wash-out rate. The AUC in this study for TI-RADS, CEUS and the combination of both were 0.806, 0.934 and 0.950, respectively [66].

A similar study was carried out by Xu et al. in which 432 thyroid nodules were analyzed with CEUS and six suspicious features were pointed out as being specific in differentiating benign from malignant nodules: slow wash-in (*p* = 0.001), slow time to peak (*p* = 0.002), non-uniform enhancement (*p* = 0.023), irregular enhancement (*p* = 0.002), unclear enhancement boundary (*p* = 0.012) and no visible ring enhancement (*p* = 0.004) [34]. As in other studies, CEUS combined with TI-RADS showed better accuracy than any of the two alone, achieving a sensitivity of 85.66% and specificity of 83.33% [49,67].

In a study conducted by Jiang et al., the diagnostic value of CEUS for thyroid nodules with calcification was analyzed and the results for differentiating benign vs. malignant disease were as follows—for conventional US sensitivity and specificity 50% and 77%, for CEUS 90% and 92%, respectively. Moreover, quantitative analysis of CEUS in thyroid nodules was performed and revealed that time to enhancement and time to peak were greater in malignant nodules than benign, but the peak intensity or malignant nodules was significantly lower than in benign nodules [36].

Hu et al. carried out a study in which CEUS quantitative analysis was used for suspicious thyroid nodules. Benign thyroid nodules showed identically in slow-out and hypo-enhancement, but malignant nodules showed a slow-in, identical-out and more hypo-enhancing appearance compared to normal thyroid parenchyma [37,50].

For a comprehensive summary of existing research on qualitative and quantitative parameters in CEUS of thyroid nodules, see Table 1.

### 3.3. Benign Thyroid Lesions

#### 3.3.1. Thyroid Adenoma

Thyroid adenoma is a benign lesion with morphologically follicular or papillary architecture [72]. Adenomas may be hormonally active causing hyperthyroidism, also known as toxic adenomas, or appear inactive [72]. Cytologic features between follicular adenoma, carcinoma and follicular variants of papillary carcinoma overlap, suggestive of the difficult differential diagnosis [73], and pathological examination is required [74]. Jiang et al. conducted a study in which thyroid adenomas were analyzed by CEUS and mainly presented with a homogenous hyperenhancement; this is due to the fact that adenomas have a complete capsule with surrounding rich blood supply [36], as they mainly grow expansively, gradually pushing the arteries and veins towards the periphery of the tumor, with the continuous growth of new capillaries. Therefore, the contrast agent reaches the center of the nodule more slowly compared with the normal surrounding tissue and wash-out is equally slow, displaying a “fast-in and slow-out” imaging pattern (Figure 1). In a study by Schleder et al. CEUS was used in a preoperative setting to differentiate thyroid adenoma vs. carcinoma in a total of 101 patients. A statistically significant difference in microcirculation between adenoma and carcinoma was noted; adenoma was characterized by no wash-out or wash-out with a persisting edge in the late phase; in contrast, thyroid carcinomas showed a complete wash-out in the late phase with CEUS sensitivity, specificity, PPV and NPV of 81%, 92%, 97% and 63%, respectively [75].

#### 3.3.2. Nodular Goiter

Multinodular goiter is characterized by an increased volume in the thyroid gland ranging from uni-nodular to multinodular and cystic enlargement of the gland [76], volume exceeding 19 mL for women and 25 mL for men as reported in a study by Teng et al. [77].

A study by Jiang et al. analyzed 62 cases of nodular goiter with CEUS and revealed predominantly homogeneous iso-enhancement due to lack of fibrous capsules and no difference in vascularization between the nodule and surrounding thyroid parenchyma, therefore the internal perfusion is similar in nodular goiters and normal thyroid. However, six cases showed inhomogeneous hypo-enhancement that could be attributed to the development pattern of nodular goiter where at the late hyperplasia stage necrosis, liquefaction and hemorrhage occur [36].

Furthermore, nodular goiter has been described as having regular high enhancing rings similar to thyroid adenoma aiding the differentiation between benign and malignant disease by using CEUS [9,75].

In a study conducted by He et al. 35 nodular goiters and 15 nodular goiters with hyperplasia were analyzed and various enhancement patterns were observed, most commonly wash-in and wash-out, similarly to thyroid parenchyma. Interestingly enough, six cases of nodular goiter in this study were misdiagnosed as malignancy due to an inhomogeneous low enhancement pattern [30].

#### 3.3.3. Thyroiditis and Lymphoma

Hashimoto’s thyroiditis is a rather common endocrine disorder and is the leading cause of hypothyroidism in iodine-sufficient parts of the world [78]. The inflammation in the thyroid gland contributes to a 40–80 times greater risk for developing primary thyroid lymphoma in comparison with patients without thyroiditis [79]. On ultrasound, Hashimoto’s thyroiditis background with a heterogeneously decreased thyroid echogenicity and hypervascularity may cause overlap between benign and malignant findings [71].

In 2015 a study on the value of CEUS was analyzed in diagnosing thyroid nodules coexisting with Hashimoto’s thyroiditis. Sixty-two nodules in a study of 48 patients were evaluated—peak intensity and enhancement pattern showed statistically significant differences between malignant and benign thyroid nodules and heterogeneous enhancement was highly suggestive of malignancy in patients with co-existing autoimmune thyroiditis with sensitivity and specificity of 97.6% and 85.7%, respectively [80].

Moreover, it is also suggested to add CEUS in subacute thyroiditis where lesions are hypoechoic with irregular margins, suggestive of malignancy, and additionally elastography data confirm suspicious stiff areas, while CEUS shows peripheral or iso-enhancement and would lead to follow-up and conservative treatment instead of surgery.

Another similar study by Yang et al. investigated the diagnostic performance of CEUS in differentiating primary thyroid lymphoma (PTL) and nodular Hashimoto’s thyroiditis (NHL) in patients with a known background of autoimmune thyroiditis. Sixty-four thyroid nodules were analyzed out of which 31 were primary thyroid lymphoma and 33 were nodular autoimmune thyroiditis. All 64 lesions presented as hypoechoic solid nodules on conventional US, but PTL was more often associated with mixed vascularity and NHT with peripheral vascularity on Doppler US. As for CEUS, most PTL lesions presented as hypo-enhanced, with the centripetal heterogeneous pattern. There were statistically significant differences between peak intensity and AUC for PTL and NHT. The diagnostic accuracy of CEUS for diagnosing PTL in patients with autoimmune thyroiditis was around 70.3–75%. The best results were accomplished if the combination of quantitative parameters, PI, TTP and AUC ratios, were used and, if combined with CEUS imaging features, showed an AUROC of 0.92 (95% CI, 0.82–0.97) [42].

In a study conducted by Wei et al. primary thyroid lymphoma was studied in 20 patients with the conventional US and 10 patients with CEUS. The conventional B mode ultrasound appearances of PTL were classified and were as follows: all cases were of hypoechoic appearance, 12 were the diffuse mass type, six of multiple nodular type and two cases of mixed type. As for CEUS, 8 out of 10 cases showed a diffuse homogenous enhancement and two cases were a diffuse heterogeneous enhancement which was linked to necrosis within the tumor. By performing quantitative analysis on CEUS parameters, TTP of the primary tumor or affected lymph nodes was longer than that of the ipsilateral common carotid artery (*p* = 0.004) [81].

### 3.4. Thyroid Cancer in CEUS 

There are several known subtypes of thyroid cancer, the most commonly described pattern of malignancy being a low enhancement on CEUS—particularly due to lack of blood supply and insufficient neovascularization, as well as interstitial fibrosis, especially in the central parts [41,82]. Qualitative patterns on CEUS suggesting malignancy include incomplete ring enhancement, heterogeneous enhancement and wash-out in the late phases; furthermore, such parameters suggestive of malignancy in the quantitative analysis include polyphasic washout curves, early arrival time and shorter TTP [54] (see Figure 2).

Hornung et al. quantitatively analyzed 22 malignant thyroid nodules with CEUS, out of which 14 were papillary, seven follicular and one medullary carcinoma. On CEUS 19 out of 22 tumors presented with a significant early arterial irregular vascularization starting at the periphery, in all 22 cases wash-out in the late arterial phase was present. AUC representing the amount of CA reaching different regions of the nodule was higher at the edge than in the center of the tumor. Interestingly, irregular peripheral vascularization by Power Doppler was only detected in 8 out of 22 patients [50]. Usually, in the literature, studies are dedicated to papillary carcinoma or are mixed studies with few other subtypes included.

#### 3.4.1. Papillary Thyroid Carcinoma (PTC)

Papillary carcinoma is the most common histological type of thyroid cancer accounting for approximately 85% and is often diagnosed in women between the third and fifth decades of life. The prognosis is excellent with a survival rate of approximately 90% at 20 years [8,83].

Papillary carcinomas morphologically are usually 2–3 cm in size on average, usually white and with an invasive appearance, characterized by a central core of fibrovascular tissue surrounded by cells with crowded oval nuclei [84].

In a study by Jiang et al. 49 papillary carcinomas were analyzed with CEUS; 44 had microcalcifications, of which 42 had inhomogeneous hypo-enhancement on CEUS [36]. A similar study by Ma et al. examined papillary carcinomas and found that a low enhancement pattern on CEUS is the most common finding for this type of tumor [52]. The reason for decreased blood supply in papillary carcinomas may be due to calcified psammoma bodies that affect tumor angiogenesis [14].

In 2015 a study was conducted by Li et al. where the performance of CEUS in diagnosing papillary thyroid microcarcinomas (<1cm in diameter) was studied. The correct diagnostic rate of CEUS was 85%, sensitivity 88% and specificity 80% for diagnosing microcarcinomas in 73 patients, but t0he use of CEUS had no clear advantages for diagnosing thyroid microcarcinomas as there were no statistically significant differences between malignant and benign nodules [61]. Furthermore, an interesting study by Gao et al. also suggests that blood-rich enhancement on CEUS is associated with a non-excellent response after thyroidectomy in a study on 306 patients with PTC [85].

A quantitative CEUS analysis of 62 patients with PTC was performed by Zhou et al., where the correlation between CEUS features and histologically determined micro-vessel density (MVD) was studied. The main peak intensity of PTC was lower than that of the surrounding thyroid parenchyma; moreover, a positive correlation was observed between peak intensity and MVD in PTC suggesting that quantitative analysis of CEUS could help determine PTC [86].

The correct staging of thyroid cancer through the identification of metastatic lymph nodes is essential for proper clinical and surgical management, for treatment planning and prognostic evaluation. Jia Zhan showed that homogeneity, cystic change or calcification, and above all intensity at peak time, were the three strongest independent predictors for malignancy in lymph nodes on CEUS [87]. In addition, benign nodes show a centrifugal progression of enhancement, while a prominent centripetal enhancement is more often observed in metastatic nodes.

#### 3.4.2. Follicular Thyroid Carcinoma (FTC)

Follicular carcinoma is the second most common thyroid malignancy classically accounting for 10–15% of all thyroid malignancies, though a decrease in incidence has been reported recently [88]. This cancer is more often diagnosed in women between the fifth and sixth decades of life. Cumulative incidence of all-cause deaths for FTC was 24% and 45% at 10 and 20 years, respectively, as showcased by Su et al. [89].

In the case of FTC surgical biopsy or excision is usually required to make the diagnosis, though over 80% of all follicular neoplasms prove to be benign [90]. It has been reported that for the detection of FTC with the predominantly internal flow, Color Doppler Ultrasound (CDUS) can be useful [91], a meta-analysis revealing that sensitivity and specificity and specificity for CDUS in predicting malignant follicular thyroid neoplasms is 85% and 86%, respectively [91,92]. Lesions showing iso-enhancement with a focal low-enhancement region should be considered malignant if the inflammatory cause has been excluded [9].

He at al. conducted a study in which only one case of FTC was analyzed with CEUS showing a fast wash-in and slow wash-out with homogenous intense enhancement without ring enhancement, similar to that of a follicular adenoma; moreover, on CDUS this lesion was rich in blood flow. This study reports that CEUS, however, cannot accurately distinguish between FTC and follicular adenoma as the only pathological diagnostic criteria for FTC was tumor invasion of the margin or blood vessels [30]. Further research on CEUS sensitivity and specificity for diagnosing FTC is required.

#### 3.4.3. Medullary Thyroid Carcinoma (MTC)

Medullary thyroid carcinomas (MTC) are rare tumors accounting for approximately 5% of all thyroid malignancies. MTC arises from parafollicular C cells of the thyroid [93] which secrete calcitonin and carcinoembryonic antigen. These are also sensitive markers in the process of MTC diagnosis, follow up and prognosis, but rare cases of calcitonin-negative MTC have been reported [94] requiring a more complex approach, possibly including CEUS. MTC can be associated with Multiple Endocrine Neoplasia type 2 (MEN2). It has to be noted that the prognosis of MTC is markedly worse than that of papillary or follicular thyroid cancer with 10-year survival being around 74% as reported by a recent study of 140 patients with MTC [95].

In a study by Zhang et al. the value of peripheral enhancement pattern for diagnosing thyroid cancer was assessed, including two cases of medullary thyroid carcinoma. One of the cases showed an irregular no-enhancement pattern that is more typical of malignant lesions, but the other carcinoma showed a regular high-enhancing ring that characterizes mostly benign lesions. The latter carcinoma was large (>5cm), heterogenous and showed a well-defined margin and rich blood flow on the conventional US, mimicking a benign lesion, which contributed to a misdiagnosis; however, such cases are sparsely reported [39].

#### 3.4.4. Anaplastic Thyroid Cancer (ATC) 

Anaplastic thyroid cancer is a rare type of thyroid tumor accounting for approximately 2–5% of all thyroid malignancies and is associated with a high mortality rate [94]. The estimated incidence is around 1–2 cases per million in a year and the peak incidence is between the sixth and seventh decades of life [96]. Histological variants of ATC include giant-cell, spindle-cell and squamoid-cell tumors. Up to 20–30% of cases morphologically show areas of necrosis and hemorrhage [96].

Only one case of ATC scanned with the conventional B mode ultrasound, CD and CEUS has been reported in the literature by Proiti et al. The diagnosis was confirmed by FNAB. On the conventional US the lesion showed a heterogeneous hypoechoic structure with irregular margins and was solitary with a maximum diameter of 3 cm. With CD no significant internal vascularity was noted, but some peripheral vessels were present. As for CEUS, a bolus of 4.8 mL of CA was used, the lesion showed an overall markedly reduced vascularity. Quantitative analysis of CEUS parameters was performed, average TTP index was 2 and average peak index was 3.4 [96]. Guisti et al. reported that a peak index of less than 1 and TTP index greater than 1 are characteristic of malignancy [49]. Additional research is required for further determination of diagnostic patterns for ATC.

For an accuracy comparison of CEUS in benign and malignant thyroid nodules, see Table 2.

### 3.5. CEUS before and after Local Treatment

Thermal ablation has been frequently applied in recent years to reduce the invasiveness of treatment in benign thyroid nodules, recurrent thyroid cancer, and metastatic cervical lymph nodes. For solid and mixed structure nodules, the most commonly used are the following—laser (LA) and radiofrequency ablation (RFA) [98,99], microwave ablation [100,101], and, lately, high-frequency ultrasound (HIFU) [102,103]. Percutaneous ethanol injection efficacy is reported based on a proportion of solid and cystic components and is effective in the treatment of predominantly cystic nodules (>90%) [104]. The primary outcome of image-guided thermal ablations was associated with a volume reduction ratio (VRR) at 6, 12, 24, and 36 months of 60%, 66%, 62%, and 53% [28].

CEUS helps to clarify boundaries between viable and nonviable tissue before and after treatment (Figure 3). This could be helpful in obtaining a more precise and reproducible measurement of the ablated area right after the ablation procedure and in the follow-up imaging—early term (3 months) and intermediate-term (6 and 12 months) are suggested intervals for follow-up with long term monitoring up to 1–2 years, to assess regrowth and to address the misinterpretation of post-treatment appearances (hypo-echogenicity), mimicking malignancy in cases of limited history data [105].

A recent systematic review reported that regrowth may be a distinct process from nodule shrinkage; furthermore, it may depend on the nodule behaviour and technical issues such as operator experience, the lack of treatment of the nodule’s margins related to the feeding artery or draining vein, and the size and the position of the nodule which influence the quality of an RFA treatment. Due to the above mentioned difficulties in performing RFA, CEUS rather than non-enhanced ultrasound may be of particular use in determining local treatment outcome in cases with thyroid nodules proximal to critical structures, as well as large thyroid nodules [106,107,108].

Contrast-enhanced ultrasound can improve ultrasound diagnostic accuracy for cervical lymph nodes staging after papillary thyroid carcinoma diagnosis. It can be useful for characterizing focal US alterations in patients with suspicions of nodal metastatic involvement, where perfusion defects are a sign of metastatic involvement: poor or absent vascularization can be identified in widespread metastatic infiltration, corresponding to areas of necrosis.

## 4. Discussion

An increase in thyroid nodule prevalence has been recorded in the last few years. High-resolution ultrasonography is the most important modality for the evaluation of a thyroid nodule. The latest guidelines have proposed criteria for risk stratification that categorizes the thyroid nodules and risk of malignancy in TIRADS systems. However, the differential diagnosis for nodules with intermediate and low suspicion is still difficult. Conventional ultrasound features such as hypo-echogenicity, irregular margins, a taller-than-wide shape, a solid internal component and microcalcification are predictive of non-follicular thyroid carcinoma. CEUS enhancement provides additional significant patterns for differentiating benign and malignant nodules with intermediate and low suspicion (5–20% malignancy risk). This risk can increase to 38.5% if the nodule shows heterogeneous enhancement [109]. With nodular goiter and subacute thyroiditis, a visible peripheral enhancement may be observed, as they are hyperplastic or atrophic in architecture. While some cases may present with low enhancement or no enhancement in subacute thyroiditis, the majority of benign nodules, including adenoma show a tendency of “fast in and slow out” enhancement, which poses a risk for misdiagnosis. There are also various meta-analyses [56,69,70], systematic reviews and original articles [35] showing controversial statements about ring enhancement as predictor of malignancy or feature of the benign nodule, leaving the community with the question as to whether CEUS in clinical practice can be associated with any additional benefit. Nevertheless, four meta-analyses [28,56,69,70] showed that both the sensitivity and specificity of CEUS were more than 85% and 82%, while PPV and NPV were 83% and 85%, respectively. Although CEUS exhibits high sensitivity and specificity, there remains a missed diagnosis rate of 12.5% and a misdiagnosis rate of 13.67% [110]. The present review showed that no isolated CEUS feature is capable of predicting thyroid malignancy with acceptable diagnostic accuracy. We have to underline that only a few papers reported on specific CEUS features of certain subtypes of the tumors such as follicular, medullary and anaplastic cancers. However, as recognized for thyroid nodule malignancy risk stratification by the US, for acceptable accuracy in malignancy a combination of several CEUS parameters should be applied: hypo-enhancement, heterogeneous, peripheral irregular enhancement in combination with internal enhancement patterns and slow wash-in and wash-out curve lower than in normal thyroid tissue. In contrast, homogeneous, intense enhancement with smooth rim enhancement, intense wash-in and slow wash-out are indicative of benignity of the thyroid nodule. In cases of indeterminate lesions by conventional US and in cases of diagnostic inconsistencies between CEUS and SWE, combined scores and further fine-needle aspiration biopsy is recommended to improve diagnostic accuracy. Furthermore, many recent studies suggest that artificial intelligence algorithms increase the accuracy of diagnosing benign versus malignant thyroid nodules, especially in the TI-RADS 4 and 5 categories, and help reduce the rate of unnecessary FNAB from 62% to 35% [111,112,113]. In 2020, Xu et al. published a meta-analysis which included 19 papers with 4781 thyroid nodules, analyzing the performance of Computer Aided Diagnosis (CAD) systems performance in differentiation of malignant thyroid nodules: the deep learning-based system showed a sensitivity of 87% and a specificity of 85%. The authors concluded that CAD systems could help, but that experienced radiologists may be superior to CAD systems, especially for real-time diagnosis [68], highlighting yet another tool that could aid differential diagnosis in the near future. 

## 5. Conclusions

In conclusion, with current further research, CEUS appears to achieve reliable performance in detecting or excluding thyroid cancer. It can also play an operative role in guiding ablation procedures on benign and malignant thyroid nodules and metastatic lymph nodes and has a promising role in the detection of extra-nodular extension in malignancy and in the evaluation of treatment response. However, there is a need for a prospective multicenter study, with a tailored approach by TIRADS categories and histology reference, to define indication and standardized qualitative techniques and parameters in order to confirm the usefulness of CEUS.

## Figures and Tables

**Figure 1 cancers-13-05469-f001:**
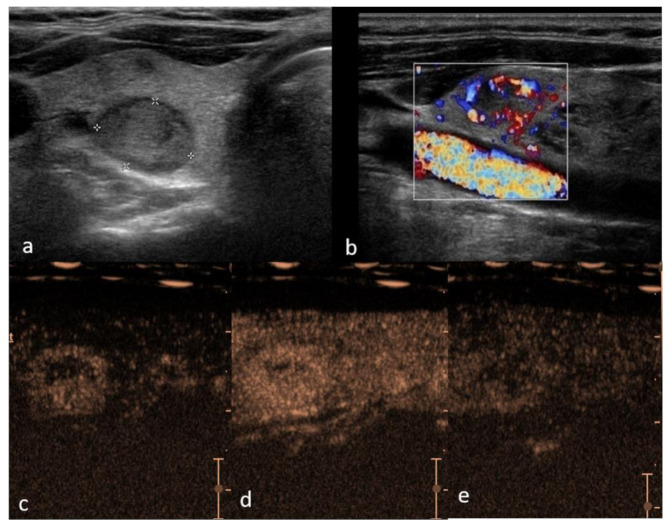
Right lobe hypoechoic lesion with halo sign, TIRADS 3, Bethesda 2, Follicular hyperplasia (**a**)—B mode hypo-echogenicity of the structure; (**b**) color Doppler shows hypervascularity in peripheral part of the lesion; (**c**) contrast enhancement is predominantly peripheral with smooth ring enhancement, with areas of rapid and intense vascularization in periphery and slow in the center (**d**) and suggestive slow wash-out (**e**) in comparison to the adjacent parenchyma.

**Figure 2 cancers-13-05469-f002:**
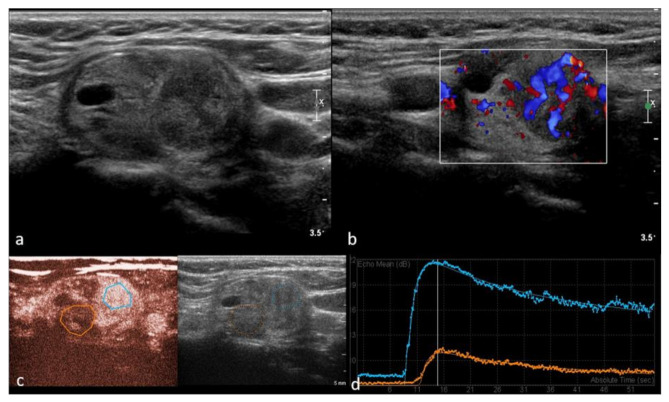
Right lobe heterogeneous lesion, TIRADS 4, Bethesda 5, Papillary cancer (**a**)—B mode hypo-echogenicity of the structure with cystic components; (**b**) Color Doppler shows hypervascularity in one part of the lesion, (**c**) contrast enhancement is heterogeneous with areas of low vascularization suggestive of malignancy and (**d**) confirming quantitative difference within the malignant tumor parts (yellow—necrotic areas, blue—intense enhancement and slow wash-out curve).

**Figure 3 cancers-13-05469-f003:**
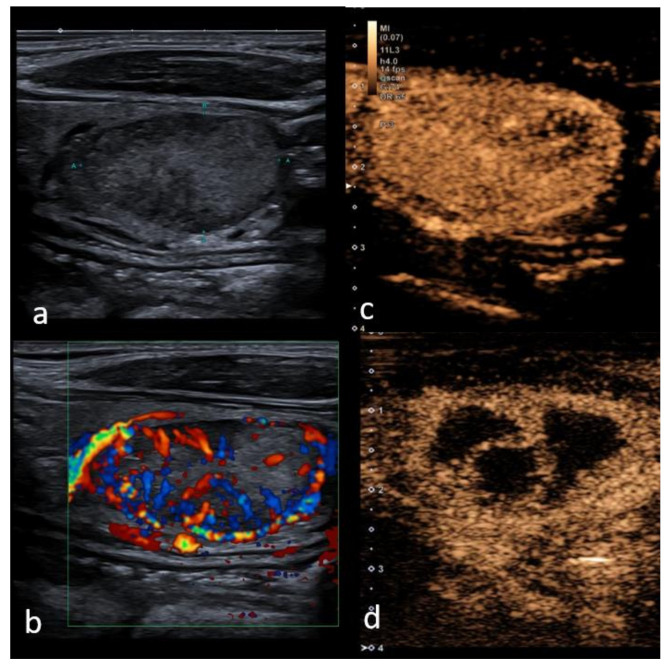
Right lobe heterogeneous lesion, TIRADS 3, Bethesda 2, Follicular adenoma, (**a***)* B mode mildly heterogeneous structure with cystic components; (**b**) Color Doppler shows hypervascularity in periphery of the lesion, (**c**) contrast enhancement is predominantly hyper-vascular and homogeneous with minor parts of lower vascularization prior to the ablation treatment; (**d**) 6 months after radiofrequency ablation volume reduction by 52% has been reached, and avascular necrotic areas (black areas) are well delineated within nodule for further treatment planning.

**Table 1 cancers-13-05469-t001:** Qualitative and quantitative CEUS parameter research summary.

Author	Year	Country	Nodules	Sensitivity	Specificity	AUC	Parameters
Zhang et al. [57]	2010	China	104	0.83	0.85	0.91	Qualitative
Nemec et al. [53]	2012	Austria	42	0.87	0.89	0.83	Quantitative
Cantisani et al. [54]	2013	Italy	53	0.78	0.83	0.87	Qualitative
Giusti et al. [49]	2013	Italy	73	0.86	0.91	0.94	Quantitative
Deng et al. [62]	2014	China	175	0.85	0.90	0.84	Qualitative
Jiang et al. [36]	2015	China	122	0.90	0.92	0.90	Quantitative
Wu et al. [55]	2016	China	229	0.95	0.95	0.77	Qualitative
Sui et al. [63]	2016	China	109	0.82	0.91	0.88	Qualitative
Zhang et al. [9]	2016	China	157	0.88	0.65	-	Qualitative
Zhang et al. [39]	2018	China	120	0.98	0.99	-	Qualitative
He et al. [30]	2018	China	88	0.79	0.95	-	Qualitative
Wang et al. [66]	2018	China	135	-	-	0.93	Qualitative/Quantitative
Zhao et al. [58]	2018	China	117	0.89	0.88	0.88	Qualitative
Xu et al. [68]	2019	China	432	0.86	0.83	0.87	Quantitative
Yang et al. [42]	2021	China	64	0.84	0.88	0.92	Quantitative
Yu et al. [69]	2014	Meta-analysis	597	0.85	0.87	0.91	-
Sun et al. [70]	2015	Meta-analysis	1154 nodules	0.88	0.90	0.94	-
Ma et al. [56]	2016	Meta-analysis	1127 patients	0.88	0.90	0.94	-
Zhang et al. [71]	2020	Meta-analysis	4827 nodules	0.87	0.83	0.93	-
Trimboli et al. [28]	2020	Meta-analysis	1515 nodules	0.85	0.82	-	-

**Table 2 cancers-13-05469-t002:** CEUS performance in benign and malignant thyroid lesions.

Benign Thyroid Lesions					
Thyroid Lesion	CEUS Characteristics	Sensitivity	Specificity	PPV	NPV
Thyroid Adenoma [74]	homogenous hyperenhancement, “fast-in and slow-out” no wash-out or wash-out with persisting edge in the late phase	0.81	0.92	0.97	0.63
Malignant thyroid lesions					
Primary thyroid lymphoma [41]	hypo-enhanced, with centripetal heterogeneous pattern, lower PI, AUC, TTP than thyroid parenchyma	0.84	0.88	0.87	0.85
Papillary thyroid carcinoma [35,60]	inhomogeneous hypo-enhancement lower PI than parenchyma	0.88–0.90	0.8–0.92	0.8	0.93
Pooled					
Benign and malignant [27,97]	histology as reference	0.85–0.88	0.82–0.90	0.83	0.85

PI—peak intensity, TTP—time to peak, AUC—area under the time–intensity curve.

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
