# Peer review of "Performance of Contrast-Enhanced Ultrasound in Thyroid Nodules: Review of Current State and Future Perspectives"

_cancers, 2021, doi:10.3390/cancers13215469_

Round 1

Reviewer 1 Report

The present is a quite complete and compehensive review on the role of CEUS in the assessment of thyroid nodule. The article is well written, with clear results and with conclusions supported by the results.
I have only minor remarks:

-          Intro: I suggest to better and clearly declare the aim of the study at thi end of this section. In my opinion the aims should be according to the results paragraphs.

-          Methods: ok

-          Results:

o   Please provide a paragraph abour complication with CEUS

o   At line 480 please expand your discussion also considering the recently published study by Cesareo et al doi: 10.3390/cancers13112746. Is there a role of CEUS to improve the efficacy of thermalablation?

-          Conclusion: please suggest which kind of study is required to achieve a solid prooof to introduce CEUS in routine clinical practice.

Reviewer 2 Report

This is an interesting and important review, including the most recent literature. The most important question is: why is CEUS – if it really is that improvement of diagnostics - not implemented in guidelines or clinical practice for the work up of thyroid nodules, although it´s available for many years now? And the answer is: CEUS is invasive, much more expensive than standard US, you need ultrasound devices which special features – which you have to buy-, it´s not approved in many countries for this indication, there is no reimbursement in most of the countries, CEUS is time consuming, you have to have a written consent of the patient in most countries, after CEUS there usually is an obligation to follow-up the patient for additional 15-30 minutes to detect allergic reaction. These are critical points in clinical practice and should be discussed in the paper. In addition, there are no studies on AI mentioned in this paper, which has been shown to improve diagnostic accuracy and reduce unnecessary FNAC. In the view of many thyrologists and thyroid associations, AI will be the future of characterization of thyroid nodules – and not CEUS. I have some minor points: - In general, self-citation of the authors seems to be quite overrepresented - Vascularisation (p2 l 74) is indeed no longer accepted as an indicator for malignancy (ATA, ETA, ACR-TIRADS, EU-TIRADS etc.), and many and most recent studies point into this direction. This should be stated more clearly. In clinical practice, many thyroid nodules are still considered to be malignant due to “hypervascularization”, although many guidelines, if not all, recommend not to use vascularization to predict malignancy. ETA, for example: R9: The routine use of Doppler US is not recommend- ed for US malignancy risk stratification. - Typo l 393: [8,79]) - L 425: this statement is not correct. Follicular carcinoma accounted for 10-20% of all thyroid carcinomas 20 years ago. Nowadays in most countries, e.g. in the US, FTC account for less than 5% of all TC – due to the increased use of ultrasound other entities, especially PTC and even MTC are more common. This should be discussed in a more balanced way - L 444 MTC: the golden standard, i.e. measuring calcitonin, should be mentioned. I can´t imagine any thyrologist, who would use CEUS instead of measuring CT. In this setting, CEUS is dispensable. Non-secretory medullary thyroid cancer might be an application for CEUS

Round 2

Reviewer 2 Report

no further comments, all points have bee  adressed adequately